**communications** engineering

# Revealing the mechanism of cold metal transfer
J. Karimi [1,2] ✉ & C. Zhao[3]

Cold metal transfer (CMT) is a pioneering feeding system widely used in wire-arc additive manufacturing (WAAM) and welding. However, process optimisation remains challenging. Although CMT has been extensively applied in various industrial sectors, its underlying mechanism is poorly understood because of the complex physics of the interactions between the wire and molten material and the wire's highly dynamic motion. To elucidate the complexity and features of CMT, we explore the dynamic behaviour and anatomy of molten materials during wire motions (withdrawal and dipping cycles) using high-speed photography at a timescale of microseconds. We reveal a crucial driving force in the melt pool and the frequent ejection of streams or particles during CMT. This study contributes to WAAM and welding by presenting the influential features of ultra-high-dynamics CMT and facilitating the progression of process optimisation.

Wire-arc additive manufacturing (WAAM) is one of the most promising additive manufacturing (AM) processes for fabricating large three-dimensional (3D) parts with a short lead time, high material efficiency, high deposition rate, and low cost. In the WAAM process, a wire is used as a feedstock to manufacture parts layer-by-layer from various metallic materials; the fabrication and post-processing times are reduced by 40–60% and 15–20%, respectively[1]. The cold metal transfer (CMT) method has been extensively used in the WAAM process to further improve the fabrication process. It requires a low heat input and high control of the process parameters and yields a smooth surface finish without spatter[2,3]. The molten pool in WAAM and arc welding is affected by different driving forces, with surface tension and electromagnetic forces being the most influential. In CMT-WAAM, the wire moves forward towards and backward from the melt pool within a few microseconds to transfer the metal with a low heat input. When the wire withdraws from the melt pool, a liquid bridge forms between the wire and the molten material before arc re-ignition[4]. The process parameters of CMT-WAAM, such as the arc voltage, slope of the power source, and circuit resistance, affect the liquid-metal bridge[4,5]. However, information regarding the interaction between the reciprocation motion of the wire and the molten materials that form the metal bridge, such as the involved force(s), is lacking. In a highly dynamic CMT-WAAM process with forward and backward wire movements within milliseconds, adjusting the bridge size and achieving optimal withdrawal and dipping cycles are challenging. Further research is required to reveal the dynamic behaviour of molten materials during CMT-WAAM. However, spatter and material ejection during CMT-WAAM have not yet been analytically investigated.

CMT is a modified feeding system used in gas-metal arc welding[6]. Researchers and engineers are continuously attempting to improve its features, and new techniques (e.g. pulse multi-control CMT with fast data processing and precise process control) have been established. Owing to its significant advantages over traditional techniques, CMT has been widely used in various industries, such as automotive, aerospace, and manufacturing[3,7]. The CMT method has been applied in welding and joining processes, particularly welding of coated thin sheets such as galvanised steels[8]. CMT is also used in surface-engineering processes, such as coating and cladding of different alloys, and offers substantial benefits over conventional welding counterparts[9]. In the CMT method, the wire exhibits a reciprocation motion in the direction of the melt pool. The forward and backward movements of the wire are generated by using a highly dynamic feeder[10]. During the forward movement of the wire, the burn arc is extinguished because of the interaction between the wire and molten pool and a reduction in the input power. The wire retracts or pulls back from the melt pool with an increase in the input power and causes arc re-ignition. The reciprocation motion of the wire, coupled with high-speed digital control, provides control over the weld geometry and energy input, along with smoother metal transfer during CMT. The current of the arc varies during the reciprocation motion of the wire and significantly reduces the heat input, which in turn results in a smaller heat-affected zone and lower dilution, distortion, and residual stress. The heat transfer and fluid flow during the CMT method have been 3D-modelled in several works. Cadiou et al.[11] estimated the forces in the melt pool during CMT, including the Lorentz forces, shear stress, arc pressure, and Joule effect for stainless steel; however, the CMT method was oversimplified. Although numerous studies on the

¹Faculty of Production Engineering, University of Bremen, Bremen, Germany. ²Institute of Materials Engineering, Technische Universität Bergakademie Freiberg, Freiberg, Germany. ³School of Materials Science and Engineering, Huazhong University of Science and Technology, Wuhan, China. ✉e-mail: jkarimi@uni-bremen.de; javadkarimimr@gmail.com

CMT method have been conducted, the metal transfer mechanism of the ultra-high-dynamics wire-feeding system has not been rigorously explored. Furthermore, although CMT has been used in various industries for over two decades, the essential features of this method are yet to be elucidated.

Recently, researchers have increased their scrutiny of nanoparticle applications in metal AM, particularly for light metals, where nanoparticles having high surface-area-to-volume ratios show unique and beneficial properties[12–14]. Nanoparticles have been employed for grain nucleation and growth during solidification to modify the microstructure, eliminate the materials' crack susceptibility, and strengthen the mechanical properties of the material subjected to AM[15–17]. Nanoparticles have considerable impact on the physical properties, particularly in additively manufactured and welded parts[18]. Several studies have reported the impact of nanoparticles on viscosity[18,19] and surface tension[20,21] and concluded that nanoparticles can intensify such properties. Here, we explore the driving force involved in the melt pool and the dynamic behaviour, anatomy, and ejection of molten materials during CMT in WAAM. We employ ceramic nanoparticles to enhance the influential material properties and verify the pioneering results. The mechanisms and features discovered in this study provide a pathway for the optimisation of the CMT process used widely in arc welding and the emerging arc AM routes.

## Results and discussion
### Essential driving force during CMT
The reciprocation motion of the wire (approximately 70 times per second[22]) plays an important role in the ultra-high-dynamics CMT method[4,23]. The wire dips in the melt pool and has physical contact with the molten materials. Here, we investigated the force involved in the melt pool during CMT and its impact on molten materials.

The CMT process parameters, such as the arc power and its slope, can control the detachment from the wire[24]. During CMT, the wire withdraws from the melt pool and forms a liquid-metal bridge with lifting-up molten materials before arc re-ignition. In a highly dynamic CMT process with forward and pullback wire movements within ~19 ms, the forces involved during the wire reciprocation motion must be analysed and quantified. The volume and weight of lifted-up materials must be quantified to determine the optimal bridge size and consequently optimise the CMT method. High-speed photography was employed to explore the forces involved in lift-up and the affected molten materials. Figure 1a shows the wire withdrawal from the melt pool for AA5183. The molten materials are lifted, showing a dark

silver colour (distinguishable from the melt pool). The lifted-up materials and melt-pool border are schematically shown (Fig. 1a). During wire withdrawal from the melt pool and before detachment, the wire lifted the molten material and formed a liquid–metal bridge. The pulling-up force for lifting molten materials plays a prominent role during WAAM or welding routes and can affect the properties of the additively manufactured or welded parts. During wire withdrawal from the melt pool, the pulling-up force, volume, and weight of the materials depend on various factors, of which surface tension and viscosity are among the most important[25]. The volume and weight of the molten materials were estimated, as shown in Fig. 1e. The pulling-up force was quantified and the values are summarised in Fig. 1f. The driving forces in the melt pools of the AM process (laser powder bed fusion (LPBF)), which play an important role in the performance of fabricated parts, were also estimated[26].

The weight and volume of the lifted-up material and the force involved during CMT were estimated; however, the obtained values had to be substantiated. Hence, we improved the material properties, such as viscosity and surface tension, to evaluate the obtained results. Ceramic nanoparticles were employed to enhance material properties because they are very small and have a very high surface-to-volume ratio. We assumed that the pulling-up force during CMT varies with the adjustment of the chemical composition and that the properties of the additively manufactured or welded parts can be influenced. Different percentages of ceramic nanoparticles were used to validate this assumption, and the volume and weight of the materials along with the pulling-up force were quantitatively examined. Three images of CMT-WAAM with different percentages of TiC nanoparticles are presented in Fig. 1b–d. With the addition of TiC, the volume of the lifted-up materials increases (Fig. 1e), as also observed in the schematics (Fig. 1b–d). Consequently, a higher pulling-up force is involved (Fig. 1f), suggesting higher viscosity and surface tension. When the material adheres to the wire, the pulling-up force lifts the molten material above the surface and partially affects the melt pool. Hence, the pulling-up force affects the melt-pool geometry in WAAM or welding. The morphology of the molten material influenced by this force is irregular and may vary with time.

In an ultra-high-dynamics CMT process, materials are transferred via highly frequent mechanical forward and pullback movements of the wire, and this reciprocation motion substantially influences the AM or welding processes. Here, we describe the driving force in the CMT feeding system. We determined the effect of this force and estimated the quantitative values of the force along with the affected material volume. The pulling-up force

**Fig. 1 | Pulling-up force in the molten pool during CMT.** Lifting materials from the melt pool for (**a**) AA5183 and AA5183 with the addition of (**b**) 2.5, (**c**) 7.5, and (**d**) 10 wt% TiC. The corresponding schematics of molten materials' interaction with the wire during withdrawal are shown below the images. The yellow solid lines denote affected (lifted-up) molten materials, and yellow dotted lines highlight the melt-pool boundaries. **e** Estimated volume and weight of the lifted-up materials. The volumes of the lifted-up materials were estimated from the zone above the orange dashed lines in the schematics. **f** Estimated force required to lift the molten materials. (Note: Error bars were calculated using standard deviation from three samples with dozens of measurements for each sample).

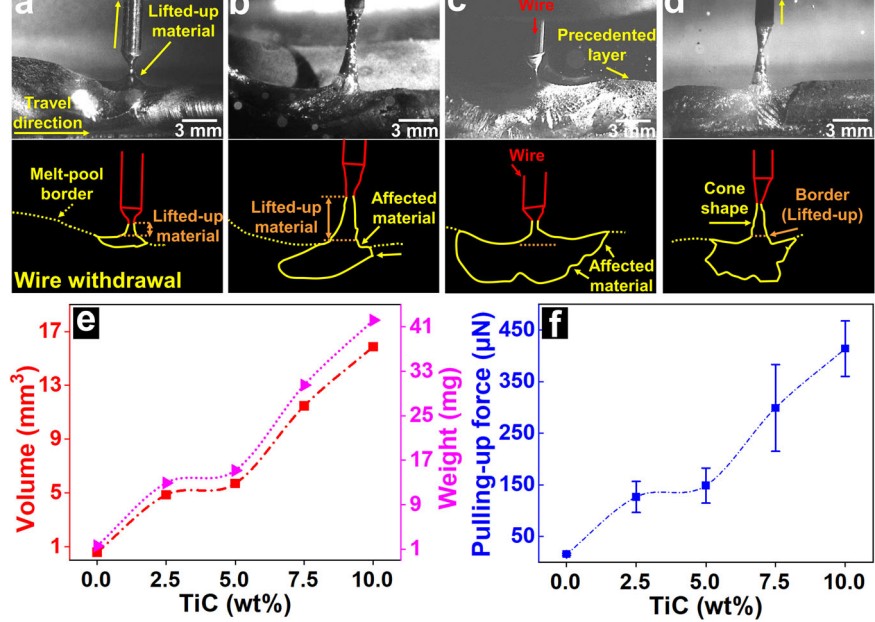

plays a pivotal role during CMT in WAAM and welding because of the reciprocation motion of the wire in the melt pools. The long-standing issues and defects of WAAM (pores, cracks, surface roughness, etc.) and the re-melting of the deposited layer are influenced by the pulling-up force, and the molten materials are partially affected by the material properties. The pulling-up force impacts the dynamics of the melt flow, where a larger volume of the molten material is affected by a higher viscosity/surface tension and the involved force magnitude. In turn, higher dynamics in the molten material could result in a higher degree of defects; for instance, it could cause a turbulent molten pool, resulting in gas or air entrapment or higher surface roughness. The force involved and its impact on molten materials and the dynamics of the CMT should be examined and properly adjusted to optimise material transfer. This is vital, particularly during the CMT-WAAM process for fabricating large parts. Process optimisation is crucial for achieving the desirable performance or properties of additively manufactured or welded parts.

### Dynamic behaviour of molten material during CMT

In the ultra-high-speed reciprocation motion of the wire during the CMT process, the withdrawal and dipping cycles should be precisely analysed to optimise the process and its features during WAAM or welding. Here, we quantitatively demonstrate the dynamic behaviour of molten materials during the reciprocation movements (withdrawal and dipping) as a function of the chemical composition.

Images of the molten material behaviour during wire withdrawal from the melt pool are shown in Fig. 2a, b, representing instances of the assessment of the dynamic behaviour depending on the chemical composition. The dynamic behaviour of molten materials affects the quality of the CMT method during metal deposition or welding[7,27]. Liquid-metal bridges or capillary bridges are formed between the molten materials and wire because of the wire reciprocation motion in molten materials with sufficiently large surface tension[10], where the bridge dimensions directly affect the withdrawal and dipping cycles. We postulate that the dynamic behaviour, including bridge dimensions and withdrawal and dipping cycles, of molten materials during CMT varies with improving material properties (viscosity and surface tension). This postulation was verified by using different percentages of TiC nanoparticles and by quantitatively examining the liquid bridge dimensions and their effects. The bridge sizes and the wire dimensions during withdrawal were measured and are depicted in Fig. 2e. The molten Al alloys with the addition of TiC nanoparticles exhibit an obvious alteration in their dynamic behaviour (Fig. 2b), showing an approximately 113% increase in the bridge size with the addition of 2.5 wt% TiC (Fig. 2e). The bridge size increases with an increasing TiC content, showing an increase of approximately 211% with the addition of 10 wt% TiC (Fig. 2e). The bridge size substantially increases with the modification of the material properties (Video S1). Accordingly, the breakup durations are measured and displayed in Fig. 2f; these increase with the addition of TiC (~51% with 10 wt% TiC).

In turn, the arc re-ignitions are postponed and the cold cycles (short-circuit arc) are prolonged. Longer cold cycles have a direct influence on the temperature field of materials during the WAAM or welding processes. The wire sizes before the breakup were measured to authenticate the measured bridge sizes, considering a constant torch distance (Fig. 2e).

Figure 2c–e indicate the wire dipping into the melt pool and the average wire lengths after arc extinctions. The size (penetration depth) of the printed/deposited layer in AM increases with the addition of TiC[15,28] because of a considerably lower thermal conductivity of TiC compared with Al alloys, the suppression of the thermal-capillary flow of materials, and an increase in heat accumulation[15]. The thermal conductivity of TiC and Al alloys is ~8 and ~237 W/mK, respectively. The height of the deposited layer changes with the addition of TiC; a lower height of the deposited layers compared with that of pure AA5183 is observed, even though the precedent layers have the same size. The wire length immediately after the arc extinction increases with the addition of TiC nanoparticles (Fig. 2e), indicating an increase in the burn-arc duration. With the wire forward movement (dipping) into the melt pool, the arc extinguishes and the WAAM or welding process reaches the cold cycle, which leads to a reduction in the heat input. Hence, the extended-arc duration extensively impacts the WAAM or welding processes. The effects of different percentages of TiC nanoparticles on the arc duration during CMT were analysed. The burn-arc durations show a noticeable extension and the arc extinguishes later compared with AA5183 (Video S2). Accordingly, burn-arc durations are measured and presented in Fig. 2f; an increase of ~53% is observed for 10% TiC addition.

The dynamic behaviour of molten materials with different chemical compositions during the withdrawal and dipping cycles in CMT was investigated, and the liquid bridge and wire sizes were quantified. Here, we describe the dynamic behaviour of molten materials using the widely used CMT methods in the emerging WAAM process and welding. The results show that the withdrawal cycles are affected by the chemical composition. In Al alloys, the bridge size during pullback is influenced by chemical composition modifications. Al has a relatively low viscosity at high temperatures (~1.3 mPa s, determined using the Arrhenius equation) and shows viscosity (at 850 °C) comparable to that of water. The TiC nanoparticles significantly increase the viscosity, thereby suppressing thermocapillary flow and increasing the stickiness of the liquid Al. Therefore, the liquid bridge size increases significantly compared with that of pure AA5183. Prolonged withdrawal reduces the temperature of the molten material, thereby influencing the temperature field. In contrast, the burn-arc duration directly influences the temperature field of materials during WAAM or welding processes, where a protracted burn-arc duration increases the heat input. In the reciprocation motion of the wire at ~52 times per second during CMT, the withdrawal and dipping cycles should be explored and tuned to optimise the process. These quantitative results provide new insights into process optimisation and the significance of the dynamic behaviour of molten materials during CMT in WAAM and welding.

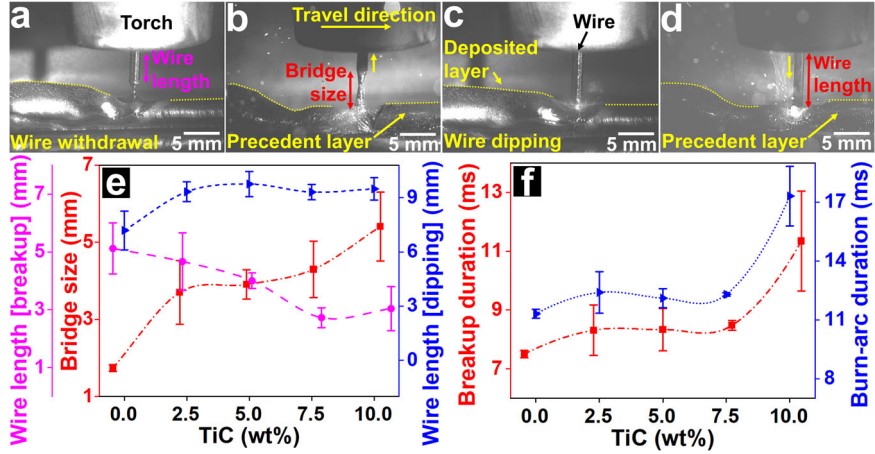

**Fig. 2 | Analysis of dynamic behaviour of molten materials.** Liquid-material bridge during wire withdrawal for (**a**) pure AA5183 and (**b**) AA5183 with 2.5 wt% of TiC. The images show the liquid bridges just (~78 µs) before the breakup (arc re-ignition). Wire dipping for (**c**) pure AA5183 and (**d**) AA5183 with 2.5 wt% of TiC. The dotted lines indicate the edges of deposited and precedent layers. Videos S1 and S2 show the withdrawal and dipping cycles, respectively, for AA5183 and with TiC addition. **e** Liquid bridge and wire dimensions during withdrawal and dipping cycles. **f** Durations of breakup (cold cycle) and burn arc (hot cycle). (Note: Error bars were calculated using standard deviation from three samples with dozens of measurements for each sample).

## Material ejection from melt pool during CMT

Manufacturers and researchers argue that CMT is a spatter-free method[8,29–31]. In this study, we investigated frequent material ejections from the melt pool. Material ejection from the melt pools was explored during wire withdrawal (cold) and dipping (hot) cycles using the ultra-high-dynamics CMT method.

Figure 3a and Video S3 indicate instances of material ejection during the wire withdrawal cycles for AA5183 when the CMT-WAAM process is in a cold cycle with no arc. Rippling rings (waves/loops on the surface) form in the melt pools after arc extinction, and their sizes increase with time. Particles are then expelled from the rippling rings (melt pools); an instance can be observed in Fig. 3a (at 750 μs). The average size of the ejected materials (particles) is found to be ~109 ± 17 μm; however, the ejected particle size cannot be precisely determined directly from high-speed photography. The materials ejected from the melt pools exhibit curved trajectories. The average velocity of particles ejected from the melt pools is found to be ~3.53 m/s. The formed rings show an average distance of ~2.3 ± 1.1 mm from the edge of melt pools and ~7.5 ± 1.3 mm from the centre of wires. The rippling rings formed during withdrawal exist for ~1.75 ms and then disappear.

During the dipping cycles, the materials are expelled from the melt pool in the presence of ignition arcs. Figure 3b and Video S3 indicate instances of streams of liquid materials ejected from the melt pool for pure AA5183 when the welding arcs do not cover the ejection zones. The ejected streams of liquids show an average distance of ~3 ± 0.6 mm from the edge of melt pools and ~7.5 ± 1.3 mm from the centre of arcs or wires. Before stream ejection, rippling rings are formed during the dipping (hot) cycles. However, they are smaller than those that appeared during the withdrawal cycles. Subsequently, the streams of molten materials are ejected from the melt pools and become fragmented. Before fragmentation, the average height of

the streams from the surface of melt pools is ~1 ± 0.3 mm. The fragmented particles (ejectum) exhibit an average velocity of ~3 ± 0.7 m/s with curved trajectories. Spatters are also observed during the arc duration and show an average velocity of ~2.2 ± 0.8 m/s. Tanaka[32] reported a spatter velocity of 6 m/s during arc welding. In the AM LPBF process, the ejection velocity has been estimated to be ~3.5 m/s[33].

Materials are frequently ejected in the shape of spherical particles or streams from melt pools during the reciprocation motion of wires with an average time of ~19 ms. The ejection of materials impacts the quality of the CMT widely used in WAAM or welding processes, which is supposed to be a spatter-free method. However, despite the large number of ejectums in the ultra-high-dynamics CMT method, the ejected materials are small and quantifying their characteristics, particularly during cold cycles, is challenging.

Ceramic nanoparticles significantly influence the characteristics and features of the molten pool of aluminium alloys in AM or welding[16,34,35]. Molten pool characteristics affect material ejection[36–38]. For instance, the melt-pool temperature influences the surface tension and, consequently, the molten metal flow, which, in turn, may affect material ejection or spattering[36]. Material ejection also depends on the geometry of the melt pools, which may be affected by melt-pool stability, energy density, and heat dissipation[37,38]. We hypothesised that the nanoparticles could influence the material ejection phenomenon during both the withdrawal and dipping cycles in CMT by considering two aspects: (1) ceramic nanoparticles can intensify the molten material properties in terms of the high surface-to-volume ratio, and (2) ceramic nanoparticles can increase the molten pool size and reduce the cooling rate. Different weight percentages of TiC nanoparticles were used to validate our hypothesis during WAAM.

During wire withdrawal cycles without burn arcs, the materials were frequently expelled from the melt pools of Al alloys with the addition of TiC

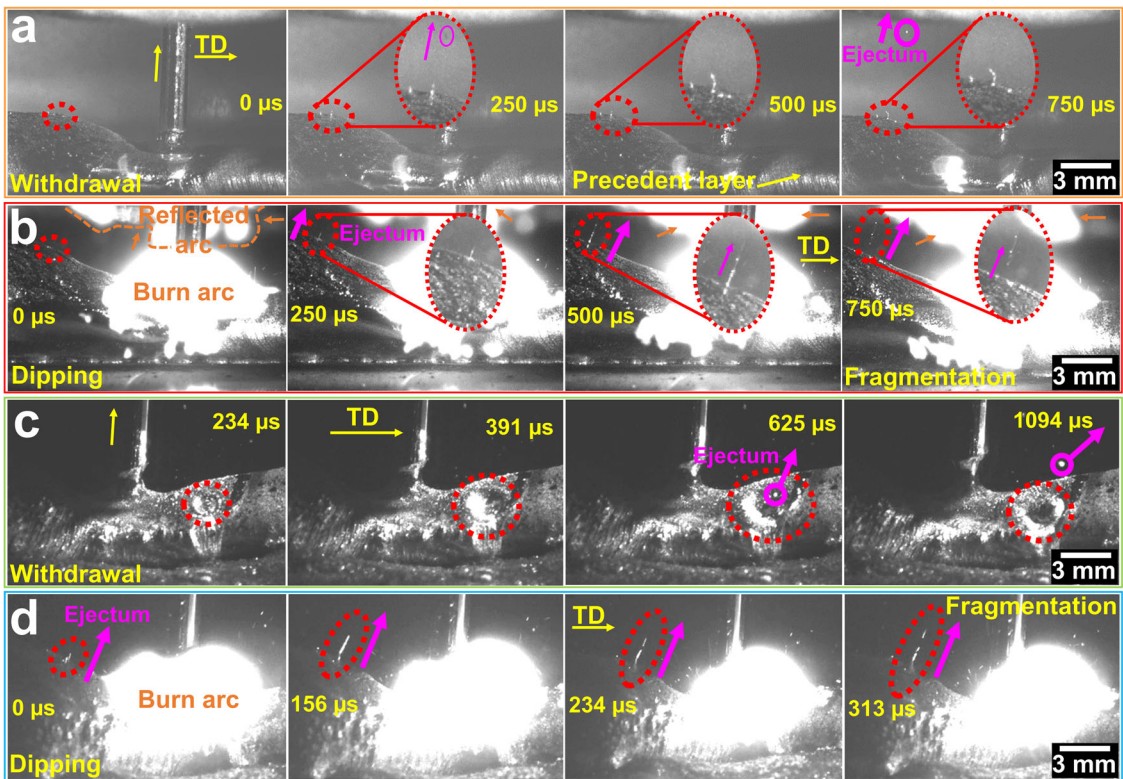

**Fig. 3 | Material ejection from the melt pools with time.** Material ejection for pure AA5183 during wire (**a**) withdrawal and (**b**) dipping. The insets depict the magnified images of material ejection. Video S3 shows the material ejection from the melt pools for pure AA5183. The purple arrows show the ejected materials from melt pools. The orange arrows highlight the arc reflection in the mirrors (located behind the wires). Material ejection for AA5183 with 7.5 wt% TiC during wire (**c**) withdrawal and (**d**) dipping. Video S4 shows material ejection from the melt pools with TiC nanoparticle addition. The recorded time (μs) of each image is shown in yellow. The material ejection in (**c**) is recorded at 234 μs because of the small size and poor visibility of the formed rippling ring (at 0 μs). 'TD' represents the travel direction of welding.

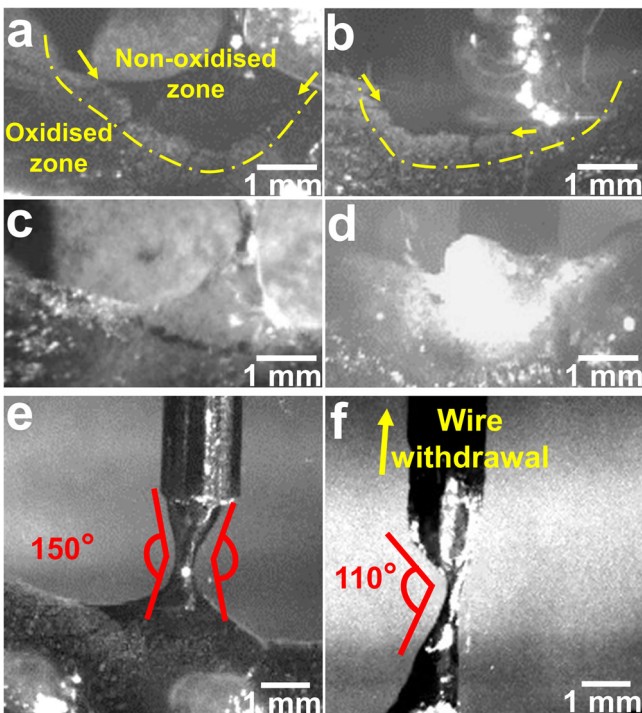

**Fig. 4 | Anatomy of deposited materials during CMT.** The oxide layer anatomy for AA5183 during (**a**) withdrawal and (**b**) dipping and for AA5183 with 2.5 wt% TiC during (**c**) withdrawal and (**d**) dipping. The dashed lines are drawn to distinguish the oxidised and non-oxidised zones, and the yellow arrows highlight their borders. The breakup shape or angle of rupture of the liquid bridge during wire withdrawal cycle from the melt pools for (**e**) pure AA5183 and (**f**) AA5183 with 10 wt% of TiC.

large for the pure Al alloys. We utilised different percentages of nanoparticles to alter the properties of the molten material and observed material ejection. The addition of TiC nanoparticles influenced the dynamic behaviour and oscillation of the molten materials and melt-pool geometries (width and depth). The observed rippling rings and ejected materials were significantly larger than those in pure AA5183 during the wire withdrawal and dipping cycles. The frequency of material ejection in CMT increased with the addition of TiC (observed in each cycle). Intensifying the material properties facilitated the quantitative analysis of the material ejection phenomenon. Our findings reveal the influence of the molten material behaviour during CMT and can lead to further advancement of this broadly used method in WAAM and welding.

## Anatomy of material deposition during CMT

The ultra-high-dynamics CMT technique significantly influences the anatomy of the deposited material. This is particularly important in quickly oxidised materials such as Al alloys. In this study, the anatomy of the deposited materials with different compositions via CMT-WAAM was investigated both analytically and numerically.

Figure 4a–d depict the interaction between the wire and molten materials during the withdrawal and dipping cycles. During CMT in AA5183 (Fig. 4a, b), two zones of different colours are easily distinguishable during the withdrawal and dipping cycles: (1) a dull grey zone and (2) a silver zone. The dull grey zone shows the non-oxidised Al, emerging around the wires during the withdrawal and dipping cycles. The silver zone indicates molten pools with the oxide layer, existing between solidified materials and zones affected by wire reciprocation motions. The dull grey zone indicates either the pinched materials (those beneath the oxide layer) or the fresh melted material via burn arc. The effects of different TiC contents on the anatomy of deposited materials were examined. The deposited Al alloys displayed a silver colour with the addition of TiC for both the pinched and fresh melted materials, and the oxidised and non-oxidised zones could not be distinguished based on the colour (Fig. 4c, d). The zone affected by wire reciprocation motions (dull grey) was not detected in samples with the addition of TiC, suggesting that the ceramic nanoparticles influenced the anatomy of deposited Al alloys. In addition, the rupture or breakup shape of the bridges in CMT was investigated; two instances are presented in Fig. 4e, f to show the angle of rupture. During the withdrawal cycle with the addition of TiC, the bridge starts to rupture from one side of the liquid materials, suggesting a high viscosity with the addition of TiC. An angle of ~110 ± 27° (for one side) is observed during the bridge rupture with the addition of TiC, as shown in Fig. 4f. By contrast, pure AA5183 shows a reduction in the diameter of liquid materials from both sides before the bridge ruptures and an angle of ~150 ± 7° for each side of the liquid materials (Fig. 4e). The difference in the angle of ruptures in the formed bridges can also be observed in Video S1.

The dynamic behaviour of the molten materials and the force involved during CMT were investigated; significant variations in these features were observed with the addition of TiC. This, in turn, influenced the anatomy of the molten pool. Four instances of melt pools with different chemical compositions were chosen to illustrate the anatomy of the deposited materials (Fig. 5a–d) during cold and hot cycles. The melt-pool geometry during the pullback movement in pure AA5183 is shown in Fig. 5a, and the results are summarised in the graph (Fig. 5e). Compared with that in pure AA5183, the melt pool width in AA5183 with TiC nanoparticles increases significantly (Fig. 5e). The melt pool width increases from ~7.9 mm in pure AA5183 to ~16 mm (over 100% wider) in AA5183 with the addition of 10 wt% TiC. The depth of the melt pool increases from ~2.7 mm in pure AA5183 to ~4 mm (~50% deeper) in AA5183 with the addition of 10 wt% TiC. During deposition, the previous layer is fully re-melted with the addition of TiC nanoparticles, whereas approximately 50% of the deposited layer is re-melted in pure AA5183. The average width and depth of melt pools in pure AA5183 show a minor deviation in the measured values (± 0.05 and ± 0.1 mm, respectively), whereas the melt pools in AA5183 with TiC additions show irregular shapes.

nanoparticles. Rippling rings formed after arc extinction, and the sizes and depths of the formed rippling rings increased significantly with time; examples are shown in Fig. 3c and Video S4. Large particles are expelled from the melt pools, and the rippling rings are closed. Materials spontaneously ejected (e.g. at ~625 μs in Fig. 3c), and the average size of the ejected materials is measured to be ~400 ± 71 μm. The materials ejected from the melt pools exhibit curved trajectories. The average ejectum velocity is found to be ~3.6 ± 0.8 m/s, similar to that for pure AA5183. The formed rippling rings exist for ~1.56 ms and then disappear.

The materials are ejected during wire dipping in the presence of ignition arcs. Figure 3d and Video S4 present instances of streams of liquid materials ejected from the melt pools, where the arcs do not cover the ejection zones. The ejected materials show an average distance of ~2.6 ± 1 mm from the edge of melt pools and ~8.2 ± 1.9 mm from the centre of the arcs or wires. Rippling rings frequently appear in the molten pool; however, they are smaller than those appearing during the withdrawal cycles. Subsequently, streams of liquid materials are ejected from the melt pools and become fragmented. Materials spontaneously ejected from the melt pools. The average height of liquid streams is observed to be ~3.1 ± 0.2 mm before fragmentation (e.g. at 156 μs in Fig. 3d). The ejected materials from the melt pools exhibit curved trajectories with an average velocity of ~5 ± 0.2 m/s.

We demonstrated material ejection in the shape of spherical particles or streams during the withdrawal and dipping cycles of CMT, which has been claimed to be a spatter-free method for metal transfer[8]. This pioneering method has been commonly assumed to improve the quality of WAAM, and it has been employed to further benefit AM for various applications. We explored the molten material behaviour and revealed material ejection during CMT-WAAM, where particles or liquid streams were repeatedly ejected from the melt pools, even when no burn arcs existed. During the wire withdrawal and dipping cycles, rippling rings formed in the molten material, and the material was ejected. However, the ejected materials were small but

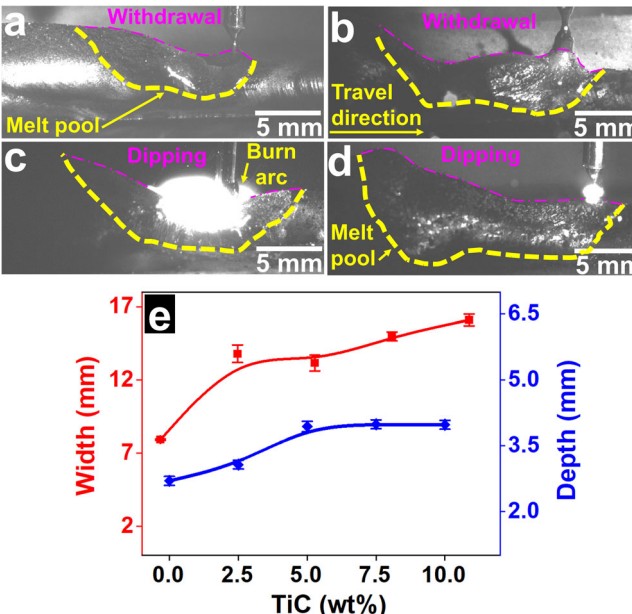

**Fig. 5 | Anatomy of melt pools during CMT-WAAM.** Melt-pool anatomy for (**a**) AA5183 and with (**b**) 2.5, (**c**) 7.5, and (**d**) 10 wt% TiC. Melt pools are shown in the cold and hot cycles. The yellow and purple dashed lines show the borders and surfaces of the melt pools, respectively. **e** Melt-pool widths and depths as functions of the chemical compositions. (Note: Error bars were calculated using standard deviation from three samples with dozens of measurements for each sample).

The size and geometry of melt pools in pure AA5183 and AA5183 with the addition of TiC nanoparticles were analysed. A distinct difference was observed because of the significant differences in thermal conductivity and heat accumulation in the samples containing TiC. In addition, the chemical composition of the CMT-WAAM parts was evaluated and showed no significant variation from the nominal composition (Supplementary Information). The average hardness of pure AA5183 and AA5183 with the addition of TiC was found to be ~81.4 ± 4.4 and ~82.3 ± 7.9 HV, respectively.

## Conclusion

In the present study, we explored the pioneering CMT method, which has been used extensively in WAAM and welding routes in various industries. The experimental and analytical results obtained by using high-speed photography revealed different aspects of the mechanism of the CMT method. To examine and verify the results, the material properties (viscosity and surface tension) were enhanced by adding different percentages of ceramic nanoparticles. We revealed the driving force in the melt pools during wire reciprocation motions and determined its effects and values. This influential force played a vital role in the melt flow dynamics in the CMT method, and the volume of the affected molten material depended on the material properties. We elucidated the dynamic behaviour of the molten materials during wire withdrawal and dipping cycles as a function of the chemical composition. These cycles adjusted by altering the chemical composition, influencing the temperature field and CMT process during WAAM or welding. We demonstrated the frequent ejection of materials in particle or stream shapes during the wire withdrawal (cold) and dipping (hot) cycles of the ultra-high-dynamics CMT method. The anatomy of the deposited materials during cold and hot cycles and the melt-pool morphology in CMT depended on the material properties. Our findings provide key insights into the pioneering CMT feeding technique that is widely used in arc welding and emerging arc AM routes. These results advance our knowledge on this feeding system and reveal CMT features that should be carefully examined and considered. Ultimately, this study paves the way for process optimisation and, in

particular, for achieving the desirable performance of large-scale WAAM parts.

## Methods

### Parts fabricated using WAAM

AA5183 (AlMg4.5Mn0.7(A)) wires having a diameter of 1.6 mm were used to fabricate WAAM components. Aluminium alloys were chosen because they are among the most commonly used materials in AM and are widely applied in various industries. A distance of 11 mm between the substrate and the contact tip of the welding torch was selected. The schematic of the CMT-WAAM process is similar to that reported by Karimi et al.[6]. Additively manufactured parts were built on an AA5083 substrate having dimensions of 200 mm × 50 mm × 20 mm. During the WAAM-CMT process, the AA5083 substrate was fixed on the working table. A standard CMT system (TPS 4000 R, Fronius, Austria), VR 7000-CMT wire feeder (Fronius, Austria), Robacta-Drive CMT PAP W Pro torch (Fronius, Austria), and Fehrenbach robot (Germany) were used to fabricate the WAAM samples. The selected CMT-WAAM process parameters are listed in Table 1. The microstructure of the wire having a diameter of 1.6 mm was investigated, and the transverse (top) and longitudinal (edge) cross sections of the wire are presented in the Supplementary Information (Fig. S1). The average hardness of the wire was found to be ~133 ± 2 HV. The chemical compositions of the wire and WAAM parts are presented in the Supplementary Information (Table S1).

### Nanoparticle dispersion in bulk samples

We fabricated AA5183 parts with TiC nanoparticles using CMT-WAAM. TiC was chosen because it is among the most important transition metal carbides and exhibits excellent intrinsic properties, particularly as nanoparticles having a high surface-to-volume ratio. TiC nanoparticles were dispersed on the substrate and/or deposited layers. TiC nanoparticles (Fig. S1) were first dispersed in polyvinyl alcohol (PVA) and then spread onto the substrate and/or deposited layers. During the fabrication process, the PVA on the substrate and deposited layers was dried before wire deposition. Note that PVA was selected because it is a non-toxic and organic material and displays excellent film-forming, emulsifying, and adhesive properties. PVA has a melting point of ~180 °C for ~88 mol.−% of hydrolysis. The viscosity of the PVA was measured using a rheological measuring device (Netzsch Kinexus Pro+), and the results are listed in Table 2. Viscosity measurements were performed at 20 °C for ~15 min. In this study, PVA having a low melting point was selected to prevent any effects on the properties of the fabricated parts. A similar method has been used to spread $TiB_2$ and TiC particles on surfaces during the WAAM of stainless steel[39]. The properties of PVA (6–88, partially saponified) are presented in Table 2. In this study, parts with different fractions of TiC nanoparticles (2.5, 5, 7.5, and 10 wt%) were additively manufactured. The chemical composition of the TiC nanoparticles was examined using energy-dispersive X-ray spectroscopy (EDX) with scanning electron microscopy (SEM), which showed ~21 wt% C and ~79 wt% Ti (close to the nominal composition). An example of the elemental mapping of CMT-WAAM for AA5183 with 10 wt% TiC is shown in the Supplementary Information (Fig. S2). Various amounts of TiC were added to the Al alloys, and those with distinguishable results are presented. During the CMT-WAAM fabrication, several samples without PVA were also fabricated to evaluate the reported results, and no significant differences were found. During the fabrication of AA5183 using CMT-WAAM, single layers with different percentages of TiC were also developed. Nevertheless, the results for the additively manufactured parts are presented herein.

### High-speed photography

High-speed photography was used to reveal the features of CMT in the WAAM process, including the forces involved in the melt pools and molten material behaviours. The experimental setup was designed, and the camera was synchronised with a computerised numerical control machine. A high-speed camera (Phantom VEO 410L, Photron, USA, Inc.) was positioned at

**Table 1 | Process parameters during CMT**

| Wire Feed Rate (m/min) | Travel Speed (mm/min) | Current (A) | Voltage (V) | Argon Flow Rate (L/min) | Layer thickness (mm) |
|---|---|---|---|---|---|
| 5.1 | 500 | ~63 | 12.5 | 20 | ~3 |

**Table 2 | Properties of PVA (4%, $H_2O$) at 20 °C**

| Material | Viscosity (mPas) | Meeting point (°C) | pH-value | Density (g/cm3) | Molar mass (g/m) | Hydrolysis degree (Mol.- %) |
|---|---|---|---|---|---|---|
| PVA | ~4.13 | 230 | 4.5–7 | 1.19 | ~37000 | 86.7–88.7 |

the front and side (perpendicular and parallel, respectively) along the travel direction of the welding torch. Images were recorded in various sequences, from 1000 to 12,800 frames per second, and the time and/or interval time between frames were mentioned in the videos and images. The high-speed camera was used to record the CMT at different frequencies and resolutions, and those with the highest quality were reported. To illuminate the WAAM process and record the CMT method, the camera was also synchronised with an illumination laser (Cavitar CaviLux HF illumination laser) with the wavelength (λ) of ~810 ± 10 nm and power of 500 W. When the CMT in WAAM was recorded, a band-pass filter having a transmissivity range of ~800 ± 20 nm was used. To increase the quality of high-speed photography, at least two mirrors were positioned around the additively manufactured parts or substrate, opposite to the camera and/or illumination laser. The reflection of light (burn arc) in the mirrors can also be observed in some images (Fig. 3b). Several videos with different setups, strategies, or cameras were recorded, and those with the highest quality and/or visibility were presented.

### Measuring the weight and volume of materials
The volume of the materials was estimated from the images obtained from high-speed photography using an image-processing software (ImageJ). Dozens of images were used to estimate the volume of the materials. These images were obtained from videos recorded parallel or perpendicular to the travel directions of the various sequences. A regular cone having a vortex at the rupture point was considered as the shape of the lifted-up materials. The volume (V) of the materials was estimated by applying Eq. 1 as follows:

$$V = \left(\frac{1}{3}\right)\pi r^2 h,\qquad(1)$$

where r and h denote the radius and height, respectively. The lifted-up material and the affected molten material as a function of the chemical composition are shown schematically in Fig. 1, and their volumes are estimated above the orange dashed line. The weights of the materials are estimated from the estimated volume and density of AA5183.

### Analysis of melt-pool force
The pulling-up force was estimated from the lifted-up molten material during wire withdrawal before rupture (arc re-ignition). The images obtained from high-speed photography were analysed using the image-processing software (ImageJ). A part of the molten material from the melt pool was lifted up, and the pulling-up force was calculated. The lifted-up parts displayed dull grey and silver colours for pure AA5183 and AA5183 with the addition of TiC, respectively. The amount of force required to lift up the material was estimated using Newton's second law. The dull grey zone in pure AA5183 had a colour different from the rest of the molten materials and was easily distinguishable. For each measurement, dozens of images were analysed using the image analysis software, and the average values were reported.

### Analysis of the dynamic behaviour of molten material
The size of the liquid-metal bridge was measured immediately before arc re-ignition. The size was measured between the tip of the wire having a 1.6 mm

diameter and the surface of the melt pool (Fig. 2b). The molten material stuck (adhered) to the wire (1.6 mm diameter) was considered as the liquid bridge, and the red arrow in Fig. 2b depicts the measured size of the liquid bridge. The wire length during bridge rupture was also measured before arc re-ignition. The length of the wire was measured from the tip of the electrode having a diameter of 1.6 mm to a point close to the welding torch. The purple arrow in Fig. 2a shows the size of the wire measured before rupture (breakup) during the withdrawal cycle. The wire size after arc extinction was also measured. The orange arrow in Fig. 2d indicates the wire length measured during the dipping cycle. The wire length was measured from the tip of the electrode to a point close to the welding torch. The distance between the torch and the surface of the substrate or deposited layer was maintained at approximately 11 mm. For all measurements, dozens of images were analysed from several videos using the ImageJ software, and the average values were reported.

Moreover, the breakup duration of the wire withdrawal from the melt pool was measured, and the time between the start of the pullback movement and the arc re-ignition was reported. The breakup duration of the cold cycle, when no arc exists, as a function of the chemical composition, is presented in Fig. 2f. The burn-arc duration was measured from the start of the arc (after breakup) to arc extinction. The time of each cycle was measured between the ruptures; a duration of 19000 ± 354 μs was observed for pure AA5183. Thus, the reciprocation movement of the wire was approximately 52.63 times per second. For each measurement, dozens of images during the cold and hot cycles from different recorded videos were analysed using the ImageJ software, and the average values were reported. All the melt pools were created using CMT with the same WAAM process parameters.

Furthermore, the viscosity of the molten materials (μ) was determined using the Arrhenius equation[40], as follows:

$$\mu = \mu_0 e^{\frac{E_\mu}{RT}},\qquad(2)$$

where $\mu_0$ is the viscosity at the reference temperature (pre-exponential factor), $E_\mu$ is the temperature coefficient of viscosity (activation energy), T is the temperature, and R is the ideal gas constant (8.314 J.mol$^{-1}$.K$^{-1}$). An activation energy of 16.114 kJ/mol[41] and a pre-exponential factor of 0.163 mPa s were used to determine the viscosity of Al at its melting point (933 K).

### Analysis of material ejection
The ejection of materials was characterised during wire dipping and withdrawal in CMT. When particle or stream ejection was observed, waves or loops appeared on the surface of the molten material. These waves or loops formed before the ejection and are referred to as 'rippling rings'. Rippling rings were observed during wire dipping and withdrawal, and their existence times were recorded. The velocity and size of the ejected streams, particles, or spatters were studied based on the travel distances. The velocities of the streams were reported after fragmentation during wire dipping. The heights of the liquid streams were measured from the surfaces of the melt pools to the tips of the streams. The materials ejected from the rippling rings were observed after ring formation, and the particles were traced across time. To measure the distance of the ejection zone from the edges of the melt pools,

the oscillation of the molten materials was observed in the videos and the edges were determined. For each measurement, dozens of images during the cold and hot cycles from different recorded videos were analysed using the ImageJ software, and the average values were reported.

## Analysis of the anatomy of the deposited materials

The oxidised and non-oxidised zones were analysed. The oxidised zones showed dull grey and silver colours for pure AA5183 and AA5183 with the addition of TiC, respectively. The dull grey colour was different from that of the rest of the molten materials and was easily distinguishable. In the samples with TiC, some small regions of non-oxidised Al were observed in the fracture zone (during breakup); however, these dimensions were not reported because of their small size and quantity. This zone can be observed in Video S1 for the sample with TiC added before rupture. In addition, the depths and widths of the melt pools were measured during the wire withdrawal and dipping cycles for comparison; however, no difference was observed between them. The melt pools were created using exactly the same process parameters and sample dimensions. In each measurement, dozens of images were analysed using ImageJ software, and the average values were reported.

## Microstructural characterisation of WAAM specimens

Microstructural analysis was performed on different areas of the specimens. The substructures of the additively manufactured Al samples were characterised by performing transmission electron microscopy (TEM, FEI TITAN 80/300) and scanning TEM (STEM). To prepare the specimens for TEM, a focused ion beam (Nova FIB) was used to extract the lamellae from the WAAM samples. The TiC nanoparticles were analysed, and the results are shown in the Supplementary Information. The STEM high-angle annular dark-field image and elemental distribution of TiC nanoparticles along with the size distribution are presented (Fig. S1). Microstructural characterisation was performed using scanning electron microscopy (Zeiss FEG) with EDX. The results are shown in the Supplementary Information. The structure of the alloys was analysed using an X-ray diffractometer (Rigaku SmartLab SE) with a D/teX Ultra 250 1D detector under Cu-Kα radiation ($\lambda = 0.154056$ nm). During the measurement, a step size of 0.01 and an exposure time of 5 s per step along the direction of deposition were used. The X-ray diffractograms are shown in the Supplementary Information (Fig. S3).

## Mechanical characterisation and density

Vickers hardness tests were conducted using a Struers DuraScan 50 G5 instrument (Vickers hardness EMCO-test). Forces of 100 and 500 g with a dwell time of 10 s were applied for the hardness measurement of the wire and CMT-WAAM samples, respectively. The density of the WAAM samples based on Archimedes' principle was measured using an analytical balance (SHIMADZU AUW120D) with a precision of $\pm 0.00001$ g. The average density of the AA5183 wire was found to be $99.31 \pm 0.036$ g/cm$^3$. The measured densities of the CMT-WAAM parts are presented in the Supplementary Information (Table S2).

## Data availability

The data that support the findings of this study are available from the corresponding author upon request.

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

## Acknowledgements
The authors acknowledge the use of facilities at the Institute of Solid-State Physics (Department of Electron Microscopy) and Inorganic Chemistry and Crystallography, University of Bremen.

## Author contributions
J. Karimi: Conceptualisation, Data curation, Formal analysis, Investigation, Methodology, Writing - original draft. C. Zhao: Investigation, Writing – review & editing.

## Funding

## Competing interests
The authors declare no competing interests.
