## [Peer review file · Communications Engineering]

Revealing the mechanism of cold metal transfer

Corresponding Author: Dr Javad Karimi

Version 0:

Reviewer comments:

Reviewer #1

(Remarks to the Author)

"Revealing the anatomy of cold metal transfer." The study is well presented; However, a few sections need improvement. The comments are following below. The work is good but in every section the English is too poor. It needs to be revised properly. Carefully revised the paper.

1. The introduction part needs improvement. The uses of CMT in different uses needs to be discussed in short "welding, coating, cladding, joining, and WAAM. Refers the papers and their related works for cladding/coating; Das, B., Panda, B.N., Sharma, F. et al. Recent Developments in Cladding and Coating Using Cold Metal Transfer Technology. J. of Materi Eng and Perform 33, 3130–3147 (2024). <https://doi.org/10.1007/s11665-023-08940-z>"
2. Revise the sentence structure. Avoid long sentences, which may be easy for readers to get the actual points. Follow in different places. For examples "During wire withdrawal from the melt pool, the pulling-up force, and volume and weight of materials depend on various factors, where surface tension and viscosity are among the most important." And "Three images of CMT-WAAM with 136 different percentages of TiC are shown in Figs. 1 b–d. With the addition of TiC, it can 137 be clearly observed that the volume of lifted-up materials increased (Fig. 1e), and 138 consequently, a higher pulling-up force is involved (Fig. 1f), suggesting higher 139 viscosity and surface tension. When materials stick to the wire, pulling-up force lifts 140 molten material above the surface and affects the melt pool partially."
3. Avoid using first person and third person sentences in the paper. For example, "Here, we quantitatively expose the dynamic behaviour of molten materials". Which needs to be "Here it is" And follow in all cases. Again "We display the dynamic behaviour of molten materials of the widely used CMT methods in the emerging WAAM process and 230 welding."
4. Figure captions should be revised. It's too long and hard to understand.
5. Avoid giving so much reference in a single place when you are discussing a single point. Put one review paper, which includes the process. Until you discuss the importances of each citing papers it's of no use. "From the beginning, CMT has been argued to be a spatter-free method by producers and researchers [26–30]".
6. Discuss their work in a small paragraph; "It has been reported that molten pool characteristics (geometry, temperature, dynamic, Etc.) affect the material ejections [34–36]."
7. Improve the sentence structure here and at many places "We show the material ejections in spherical particle or stream shapes that occurred both during the withdrawal and dipping cycles of CMT, which has been claimed to be a spatter-free method to transfer materials."
8. Who will note it "Note that yellow and 395 purple dashed lines show the borders and surfaces of the melt pools, respectively." Revised the English at different sections.
9. What does authors means by this "A Vickers hardness test (DuraScan 611 50G5, Struers, Emco-Test) was used for the hardness tests."

Reviewer #2

(Remarks to the Author)

This work studied the anatomy of cold metal transfer. The present research contributes to WAAM and welding by uncovering the influential features of ultra-dynamic CMT. However, several key issues are not clearly clarified including the following points:

1. How is the pulling-up force calculated? Please provide detailed instructions. It would be better to add an error bar in Fig. 1f.
2. Please provide more photos to explain the materials ejections from the melt pools in Fig. 3.
3. The paper should contain the conclusion. Please highlight the key findings, the scientific value, and the contribution of the current work.

4. In fig. S2, TiC is not well characterized. Can you provide a better element mapping to prove where TiC is?

Reviewer #3

(Remarks to the Author)

The present work was aimed to explore the metal transfer conditions in cold metal transfer-based wire arc additive manufacturing (WAAM). The subject under analysis is very interesting and presents technical and scientific relevance. However, the paper has a huge weakness, which is related to the way it is structured. It avoids a proper understanding of the work carried out by the authors and the results/conclusions they achieved. So, the reviewers think this paper shall be rejected and resubmitted. In this current form, it is very difficult to have an idea of the scientific merits of the paper because the paper's structure avoids its good understanding.

Please, consider the following comments:

- 1) The quality of the text must be improved. An overall improvement in the text is required. An effort must be made to standardise terms used throughout the manuscript and figures.
- 2) The Introduction section is extensive, but the objective of this work is not clearly referred by the authors. They must refer it in a very explicit way in the Introduction.
- 3) The Results and discussion's section is presented before the Materials and Methods' section. This totally avoids a proper understanding of the research. The results are not supported by a previous section on the experimental procedure.
- 4) For readers to fully understand the paper, an explanation and conceptualisation of some concepts should be provided in the text, as well as their influence and impact on the process. Here are a few situations: rippling rings and their formation, and how nanoparticles addition influences the properties of the deposited materials.
- 5) Many results were included as supplementary material to the manuscript. This makes it difficult to understand the paper. It would be better to include these results as figures in the paper so that a simpler correlation with the text could exist. The authors included 4 figures and 2 tables, which is below the maximum number of display items allowed by the journal (10).
- 6) The manuscript does not contain a Conclusions section.

As the results seem to be interesting and a deep experimental procedure was followed, these aspects must be addressed by the authors so that the scientific merits of the paper can be analysed by the reviewers.

Reviewer #4

(Remarks to the Author)

Submitted as a co-review: I co-reviewed this manuscript with one of the reviewers who provided the listed reports. This is part of the Communications Engineering initiative to facilitate training in peer review and to provide appropriate recognition for Early Career Researchers who co-review manuscripts.

Version 1:

Reviewer comments:

Reviewer #1

(Remarks to the Author)

The revised manuscript has improved in quality

Reviewer #2

(Remarks to the Author)

The manuscript has been revised and is now acceptable.

Reviewer #3

(Remarks to the Author)

Based on the response of the authors to the reviewer comments and the changes implemented in the manuscript, the paper can be accepted for publication.

Reviewer #4

(Remarks to the Author)

Submitted as a co-review: I co-reviewed this manuscript with one of the reviewers who already provided an answer.

The authors sincerely thank the reviewers for taking the time to read the manuscript. We have answered all the queries raised and modified the manuscript based on the comments, suggestions, and positive criticism. We believe that the quality of the revised version has improved substantially. The modifications are suitably highlighted in the manuscript.

Reviewers' comments:

Reviewer #1 (Remarks to the Author):

“Revealing the anatomy of cold metal transfer.” The study is well presented; However, a few sections need improvement. The comments are following below. The work is good but in every section the English is too poor. It needs to be revised properly. Carefully revised the paper.

We thank Reviewer #1 for the comments and suggestions that helped us to improve the quality of the manuscript. The authors carefully revised the introduction and body of the manuscript. The writing of the manuscript was rechecked/corrected by language professionals. The coherency, quality and readability of the manuscript are also improved.

1. The introduction part needs improvement. The uses of CMT in different uses needs to be discussed in short “welding, coating, cladding, joining, and WAAM. Refers the papers and their related works for cladding/coating; Das, B., Panda, B.N., Sharma, F. et al. Recent Developments in Cladding and Coating Using Cold Metal Transfer Technology. J. of Materi Eng and Perform 33, 3130–3147 (2024). <https://doi.org/10.1007/s11665-023-08940-z>”

We thank the reviewer for this comment. We have improved the introduction part significantly. As suggested, the different uses of the CMT method in the different processes are discussed briefly, and related papers and works are cited in the manuscript. In addition, more information was added in the introduction part.

2. Revise the sentence structure. Avoid long sentences, which may be easy for readers to get the actual points. Follow in different places. For examples “During wire withdrawal from the melt pool, the pulling-up force, and volume and weight of materials depend on various factors, where surface tension and viscosity are among the most important.” And “Three images of CMT-WAAM with 136 different percentages of TiC are shown in Figs. 1 b–d. With the addition of TiC, it can 137 be clearly observed that the volume of lifted-up materials increased (Fig. 1e), and 138 consequently, a higher pulling-up force is involved (Fig. 1f), suggesting higher

139 viscosity and surface tension. When materials stick to the wire, pulling-up force lifts 140 molten material above the surface and affects the melt pool partially.”

The sentence structure in the introduction and body of the manuscript was modified. Long sentences were either eliminated or split. Many sentences were rewritten to improve readability and followability.

3. Avoid using first person and third person sentences in the paper. For example, “Here, we quantitatively expose the dynamic behavior of molten materials”. Which needs to be “Here it is” And follow in all cases. Again “We display the dynamic behavior of molten materials of the widely used CMT methods in the emerging WAAM process and 230 welding.”

The authors thank Reviewer #1 for this comment. The authors are encouraged the active voice in the writing to highlight the ownership of the data. We retain first-person pronouns because the target journal does not prohibit these and the editor has encouraged the use of an active voice.

4. Figure captions should be revised. It’s too long and hard to understand.

Figure captions were revised to make them easier to understand and follow. However, the information in the captions is necessary to understand the images, where the journal notes that “*Legends should be detailed enough so that each figure can be understood in isolation from*”. Nevertheless, the captions’ lengths are below the maximum words (350) allowed by J. of CommsEng.

5. Avoid giving so much reference in a single place when you are discussing a single point. Put one review paper, which includes the process. Until you discuss the importances of each citing papers it’s of no use. “From the beginning, CMT has been argued to be a spatter-free method by producers and researchers [26–30]”.

As suggested, the authors have revised the citations in the manuscript, and reduced the references, particularly for the single-point discussion/explanations. However, in some cases, we added a few references to report a statement supported by the producers, researchers, and engineers. In the mentioned sentence with 4 references, we cited a dominant producer, a review paper on CMT, and the research papers on the characterization of CMT and welding(/cladding) process(es). Here, we cited descriptions, reports, and research articles to have sufficient supporting information for the statement.

6. Discuss their work in a small paragraph; “It has been reported that molten pool characteristics (geometry, temperature, dynamic, Etc.) affect the material ejections [34–36].”

The authors thank the Reviewer #1 for this comment. We added more information regarding this comment in the revised manuscript. The authors decided to add the information in the existing paragraph.

7. Improve the sentence structure here and at many places “We show the material ejections in spherical particle or stream shapes that occurred both during the withdrawal and dipping cycles of CMT, which has been claimed to be a spatter-free method to transfer materials.”

We have revised the sentence structure in the introduction and body of the manuscript. Most of the sentences were re-written, and the manuscript was scanned carefully to eliminate any typos and grammatical errors.

8. Who will note it “Note that yellow and 395 purple dashed lines show the borders and surfaces of the melt pools, respectively.” Revised the English at different sections.

The manuscript was revised carefully to eliminate the errors and improve the language. Also, the manuscript was grammatically re-checked and corrected.

9. What does authors means by this “A Vickers hardness test (DuraScan 611 50G5, Struers, Emco-Test) was used for the hardness tests.”

The sentence was rewritten to avoid any doubt. The *Materials and Methods* section was re-checked and modified. More information about the fabrication process, tests, and analysis were added.

.....

Reviewer #2 (Remarks to the Author):

This work studied the anatomy of cold metal transfer. The present research contributes to WAAM and welding by uncovering the influential features of ultra-dynamic CMT. However, several key issues are not clearly clarified including the following points:

1.How is the pulling-up force calculated? Please provide detailed instructions. It would be better to add an error bar in Fig. 1f.

We thank the Reviewer #2 for the comments and suggestions. We explained the method for calculation of pulling-up force in “*Analysis of melt pool force*” part page 25. This section was improved and more details and explanations were added. As suggested an error bar was added to Fig. 1f. In addition, all figures and tables were modified, and new images were introduced.

2. Please provide more photos to explain the materials ejections from the melt pools in Fig. 3.

We recorded the CMT process during wire withdrawal and dipping from 1000 to 20000 frames per second for the materials with different percentages of TiC nanoparticles, meaning 0, 2.5, 5, 7.5, and 10 wt%. The CMT method was recorded with many setups, material compositions, and nanoparticle addition strategies. The material ejection can be observed from the videos (Videos S3-S4), where the videos were recorded with high frames per second and showed multiple ejections for each composition in withdrawal and dipping cycles. It has been observed that different content of TiC nanoparticles showed similar material ejections, and no significant difference was observed.

In Fig. 3, the images were obtained from the high-speed photography software, showing material ejection during wire withdrawal and dipping for 0 and 7.5 wt% of TiC nanoparticles. We analysed the material ejection in videos for all compositions and the average values were reported in the manuscript. The authors decided to present the images of pure AA5183 and AA5183 with 7.5 wt% TiC to prevent repetition. More images of the material ejection during wire withdrawal and dipping for other compositions are shown in Fig. R1-A (below). The authors added more details to the “Material ejection from melt pool during CMT” part and improved this part for better readability and coherency. In addition, the images in the manuscript were modified to improve their quality.

Fig. R1-A. Image of material ejection with the addition of TiC nanoparticles. The orange and red arrows show the ejection and its direction, respectively.

3. The paper should contain the conclusion. Please highlight the key findings, the scientific value, and the contribution of the current work.

Thank you for this comment. As suggested, the heading for the conclusion part was added. The key findings, the scientific value, and the contribution of the current work were clearly explained in the conclusion.

4. In fig. S2, TiC is not well characterized. Can you provide a better element mapping to prove where TiC is?

The authors thank the reviewer #2 for this comment. We have used different characterization methods, (such as XRD, SEM, and TEM) to reveal the homogeneous (/inhomogeneous) elemental distribution, and/or microstructural properties. In particular, TEM was used to investigate TiC nanoparticle distribution in AM parts using WAAM, where the samples were prepared using FIB with a dimension (lamellae size) of ~20 μm . We observed some spherical shapes on the surface of the lamellae in the range of nanometers (~20 – ~80 nm) that could be TiC nanoparticles. However, the EDS in TEM could not confirm the chemical composition. Meaning, that the obtained chemical composition did not match the nominal chemical composition of TiC. The TEM images, including HAADF and mapping, are shown in Fig. R1-B and R1-C, respectively. Therefore, we decided to not add the TEM result to the present manuscript to prevent any future doubt. We assumed that using FIB for cutting the lamellae could affect the traceability of nanoparticles. On the other hand, we had a limitation for the TEM sample preparation method (i.e., other TEM sample preparation methods), and we had a limited budget for TEM observation since it is not available in our institute for further investigation. Hence, we just added EDS/SEM elemental mapping in the appendix. Nevertheless, the XRD results show the presence of TiC nanoparticles in the fabricated WAAM parts.

Fig. R1-B. TEM images of WAAM with the addition of TiC nanoparticles. EDS/TEM shows the chemical composition of the selected areas shown in high-angle annular dark-field image (HAADF).

Figure R1-C. One example of the elemental mapping of WAAM AA5183 with the addition of TiC using TEM.

In the present research, we showed the features of CMT and explored the dynamic behavior, and anatomy of molten materials during the withdrawal and dipping cycles. We showed that chemical composition modifications (the addition of TiC) cause a significant impact on the

features of ultra-dynamic CMT. The obtained results of CMT AA5183 were verified with those of adjusted chemical compositions.

.....
Reviewer #3 (Remarks to the Author):

The present work was aimed to explore the metal transfer conditions in cold metal transfer-based wire arc additive manufacturing (WAAM). The subject under analysis is very interesting and presents technical and scientific relevance. However, the paper has a huge weakness, which is related to the way it is structured. It avoids a proper understanding of the work carried out by the authors and the results/conclusions they achieved. So, the reviewers think this paper shall be rejected and resubmitted. In this current form, it is very difficult to have an idea of the scientific merits of the paper because the paper's structure avoids its good understanding.

The authors thank the Reviewer #3 for the comments and suggestions. The structure of the manuscript was revised. The sentences' structures were edited, and many parts of the introduction and body of the manuscript were rewritten. The written language of the introduction and body of the manuscript was modified by language professionals. In addition, we improved the precision and concision of writing language, organization (E.g., headings), sentence structure, and descriptions of the introduction and body of the manuscript. We modified the manuscript extensively, and tried to clearly show the findings and articulate the gaps in current research. We believe that readability, followability clarity and coherency are improved significantly.

Please, consider the following comments:

1) The quality of the text must be improved. An overall improvement in the text is required. An effort must be made to standardise terms used throughout the manuscript and figures.

We have improved significantly the quality of the text. The authors carefully scanned the manuscript to eliminate errors/typos. The writing language was revised by language professionals to transfer clearly the findings. The "*Materials and Methods*" section (data analysis techniques) was modified to describe this part clearly and in detail. We tried to re-write the manuscript to have a well-written and organized, with a clear and logical presentation of the introduction, results and discussion, and conclusion. More information was introduced to the terms used in the manuscript. In the revised manuscript, the terms were clearly explained and standardized in the introduction, body (of the manuscript), and materials and methods.

2) The Introduction section is extensive, but the objective of this work is not clearly referred by the authors. They must refer it in a very explicit way in the Introduction.

We thank the Reviewer #3 for this comment. The introduction section was modified and the gaps were highlighted. The motivations/objectives (particularly in the last paragraph) in the introduction part were modified to explicitly express the motivation/objective of this work. In addition, further details were added to explain the aims of the present work.

3) The Results and discussion's section is presented before the Materials and Methods' section. This totally avoids a proper understanding of the research. The results are not supported by a previous section on the experimental procedure.

The authors appreciate the reviewer's comments and suggestions. Regarding the order and structure, we were informed that the Journal of Communications Engineering appreciates both approaches: when the "*Materials and Methods*" section is placed before and after the "*Results and discussion*" section. Here, we decided to place the "*Materials and Methods*" section after the results. In addition, we improved extensively the "*Materials and Methods*" section and more details were added to support the *Results* section, and improve the readability.

4) For readers to fully understand the paper, an explanation and conceptualisation of some concepts should be provided in the text, as well as their influence and impact on the process. Here are a few situations: rippling rings and their formation, and how nanoparticles addition influences the properties of the deposited materials.

As suggested, more/new details for the concepts were added/introduced in the manuscript. In addition, we tried to clearly explain the causes or impacts of the phenomena during the process. In particular, the term "rippling rings" was further explained on page 27.

Regarding the rippling rings and their formation in the CMT process, let us briefly explain here. The rippling rings were formed in the melt pools, and then particles/streams of molten materials were ejected. Based on our observation, the rippling ring were formed in the different regions of the melt pools, i.e., in the front, rear, top, bottom, or near the burn arc. We showed the rippling rings formation and the material ejections shapes and features. However, some aspects of the phenomena are unknown to the authors and need further investigation. New experimental setups, such as beamline investigation, are needed to reveal the influence(s) as well as the cause(s) of rippling ring formation during the CMT process. These investigations are needed to trace to source/nature of the ejected particles and probably trace these particles inside the melt pool and after ejections. The beam line experimental method is not available in our research institute and such experiments could not be carried out by the authors.

However, we tried to explain the phenomena, and the manuscript was modified to provide further details in the revised version.

The explanation of the influences of nanoparticle addition on the properties of deposited materials was improved and new details were introduced. In addition, we improved the explanation of the TiC nanoparticle impacts on the dynamic behavior and the involved forces during CMT. We tried to address these investigations in the manuscript comparably and comprehensively.

5) Many results were included as supplementary material to the manuscript. This makes it difficult to understand the paper. It would be better to include these results as figures in the paper so that a simpler correlation with the text could exist. The authors included 4 figures and 2 tables, which is below the maximum number of display items allowed by the journal (10).

We agree with the reviewer that the body of the manuscript could include some of the results from supplementary material to have better accessibility. As suggested, some results from the supplementary material were introduced in the manuscript. The author changed the figures, and the number of figures increased. Some explanations were also added to the manuscript from supplementary material. However, we have prepared a manuscript with the main text just below the maximum word number (5000 words) allowed by the journal of CommsEng. Nevertheless, the authors think that the introduction of some other images from the supplementary file in the body of the manuscript may reduce the coherency; and the information is not part of the main messages of the manuscript. Furthermore, we could have more images or add similar images with different conditions (E.g., different chemical compositions) of the process. For instance, EDS or elemental mapping of one composition are shown in the supplementary material to provide supporting information. But, if we introduce elemental mapping in the manuscript, we may need to show for all of the compositions to keep consistency of the structure. However, it will be more than the maximum word numbers allowed by the journal.

6) The manuscript does not contain a Conclusions section.

Thank you for this comment. As suggested, the heading for the conclusion part was added and key findings and contributions of the current work were clearly explained in the *Conclusion* section.

As the results seem to be interesting and a deep experimental procedure was followed, these aspects must be addressed by the authors so that the scientific merits of the paper can be analyzed by the reviewers.

The authors thank Reviewer #3 for the time reviewing our paper. We improved the quality and readability of the manuscript based on the comments. The structure of the manuscript was modified, and the findings were clearly explained. We re-wrote many parts of the manuscript to improve the precision and concision of language. All the authors re-check several times the manuscript to improve its readability, followability, clarity, and coherency, where the scientific merits of the manuscript could be analyzed.

.....

Reviewer #4 (Remarks to the Author):

Submitted as a co-review: I co-reviewed this manuscript with one of the reviewers who provided the listed reports. This is part of the Communications Engineering initiative to facilitate training in peer review and to provide appropriate recognition for Early Career Researchers who co-review manuscripts.

The authors thank Reviewer #4 for the comment(s) and suggestion(s), that help us to improve the quality of the manuscript.

We thank the reviewers for their time. The authors are happy that reviewers are satisfactory to the modification of the manuscript. We have revised the manuscript to comply with the journal policies and formatting style.

Reviewers' comments:

Reviewer #1 (Remarks to the Author):

The revised manuscript has improved in quality

We thank Reviewer #1 for the comments and suggestions.

.....

Reviewer #2 (Remarks to the Author):

The manuscript has been revised and is now acceptable.

We thank the Reviewer #2 for the comments and suggestions.

.....

Reviewer #3 (Remarks to the Author):

Based on the response of the authors to the reviewer comments and the changes implemented in the manuscript, the paper can be accepted for publication.

The authors thank the Reviewer #3 for the comments and suggestions.

.....

Reviewer #4 (Remarks to the Author):

Submitted as a co-review: I co-reviewed this manuscript with one of the reviewers who already provided an answer.

The authors thank Reviewer #4 for the comment(s) and suggestion(s).